# Acid phosphatase-like proteins, a biogenic amine and leukotriene-binding salivary protein family from the flea *Xenopsylla cheopis*

Stephen Lu [1✉], John F. Andersen [1], Christopher F. Bosio[2], B. Joseph Hinnebusch[2] & José M. Ribeiro[1]

The salivary glands of hematophagous arthropods contain pharmacologically active molecules that interfere with host hemostasis and immune responses, favoring blood acquisition and pathogen transmission. Exploration of the salivary gland composition of the rat flea, *Xenopsylla cheopis*, revealed several abundant acid phosphatase-like proteins whose sequences lacked one or two of their presumed catalytic residues. In this study, we undertook a comprehensive characterization of the tree most abundant *X. cheopis* salivary acid phosphatase-like proteins. Our findings indicate that the three recombinant proteins lacked the anticipated catalytic activity and instead, displayed the ability to bind different biogenic amines and leukotrienes with high affinity. Moreover, X-ray crystallography data from the XcAP-1 complexed with serotonin revealed insights into their binding mechanisms.

[1] Laboratory of Malaria and Vector Research, National Institute of Allergy and Infectious Diseases, Bethesda, MD, USA. [2] Laboratory of Bacteriology, National Institute of Allergy and Infectious Diseases, Hamilton, MT, USA. ✉email: Stephen.lu@nih.gov

Blood-seeking behavior is believed to have independently evolved more than 20 times within the Arthropoda phylum[1]. To successfully acquire blood, hematophagous vectors must overcome various host defensive responses, such as hemostasis and inflammation. To accomplish this, they have developed complex and distinct salivary mixtures with rich pharmacological activity that interferes with blood clotting, platelet aggregation, vasoconstriction, and modulates the host immune response at the feeding site[1]. Exploration of salivary glands and saliva from hematophagous vectors in recent decades has revealed a remarkable degree of convergent evolution among salivary proteins. Specifically, distinct protein families have independently evolved to possess a common activity. One such example is the presence of proteins that exhibit high affinity binding to biogenic amines and eicosanoids. Currently, salivary proteins with this activity have been identified in mosquitoes (D7)[2], sand flies (yellow protein)[3], horse flies (antigen-5)[4], ticks, and kissing bugs (lipocalins)[5–7]. These protein families are often among the most abundant salivary proteins and are commonly referred to as kratagonists, combining the Greek words "krato" (meaning "hold") and "agonists"[8].

*Xenopsylla cheopis*, a common rat flea, is the primary vector of the Gram-negative bacterium *Yersinia pestis*, which causes the bubonic plague. This devastating disease has played a significant role in human history and continues to pose a burden in modern times[9–11]. Fleas are also implicated in the transmission of other medically and veterinary relevant pathogens, including *Rickettsia typhi* (murine typhus), *R. felis* (flea-borne spotted fever), and *Bartonella henselae* (cat-scratch disease)[12,13]. Despite their significant medical and economic importance, only a few flea salivary proteins have been functionally characterized[14–17], and the pharmacological activity of flea salivary glands remains largely unknown. Exploration of the flea salivary gland using transcriptomics and proteomics approaches has revealed its unique composition with an abundance of several acid phosphatase-like sequences lacking one or two catalytic residues[18,19].

Enzymes are systematically categorized into families based on their molecular structural archetype, catalytic residues, and the specific reactions they catalyze[20]. For example, the histidine phosphatase superfamily encompasses a functionally diverse group of proteins. However, all members share a common catalytic core based on two histidine residues[21] that undergo phosphorylation during the hydrolysis of phosphate monoesters[22]. Based on their sequence similarity, members of this family are currently divided into two branches, represented by the PFAM entries PF00300 and PF00328[23]. The first branch primarily comprises intracellular bacterial proteins with a wide spectrum of functions, while the second branch predominantly features extracellular eukaryotic proteins, specifically acid phosphatases and phytases. Notably, members of the acid phosphatase family exhibit optimal enzymatic activity at an acidic pH and can be further categorized based on their sensitivity to L(+)-tartrate[24]. Those that resist this inhibitor are commonly referred to as tartrate-resistant acid phosphatases. Interestingly, an increasing number of sequences have been discovered, which exhibit a conserved structural archetype of an enzyme family but lack one or more of their catalytic residues. In the case of the acid phosphatase-like sequences found in *X. cheopis* salivary glands, it has been proposed that the absence of catalytic activity may result in the protein being permanently bound to its substrate, potentially functioning as a kratagonist[25].

In this study, we conducted a comprehensive characterization of the structure and function of the three most abundant acid phosphatase-like transcripts identified in the salivary gland of the rat flea *X. cheopis* (XcAP-1, XcAP-2, and XcAP-3). Our findings confirmed that these three recombinant proteins lack the expected catalytic activity associated with the acid phosphatase family. Additionally, XcAP-1, XcAP-2, and XcAP-3 exhibit high affinity binding to biogenic amines and leukotrienes. Finally, the crystal structure of XcAP-1 in complex with serotonin, provided an insight into the mechanism by which acid phosphatase-like proteins from *X. cheopis* salivary glands act as kratagonists.

## Results and discussion

**The presence of acid phosphatase-like sequences lacking their catalytic residues is a distinctive characteristic observed solely in fleas.** Through a comparative analysis of *X. cheopis* salivary acid phosphatase-like sequences with previously cataloged sequences from various blood-feeding vectors, including fleas, mosquitoes, sand flies, ticks, triatomines, biting flies, and mites, it was observed that only sequences from fleas exhibited mutations in one or both catalytic residues (Supplemental file 1). Moreover, the phylogenetic analysis of these acid phosphatase-like sequences revealed the formation of a distinct clade specifically represented by sequences from fleas (*X. cheopis* and *C. felis*) (Fig. S1). Notably, within this clade, a diverse set of sequences containing both, one, or none of the catalytic histidine residues was identified. This observation suggests that the substitutions of the catalytic residues represent recent evolutionary events, restricted to the Siphonaptera order. Additionally, a comparison between the sequences coding for acid phosphatase-like sequences from the recent *C. felis* genome[26] and previous sialome studies[19,27] revealed a higher number of transcripts in the transcriptome studies. This discrepancy may be attributed to the considerable genomic plasticity observed in fleas (*C. felis* and *X. cheopis*), where individuals from the same colony exhibit significant variation in genome size[26]. These findings collectively support the hypothesis that flea acid phosphatase-like sequences occupy a genomic region that is susceptible to rapid evolution. Furthermore, it is likely that these mutated transcripts arose from catalytically active acid phosphatase sequences through the process of gene duplication and subsequent evolutionary adaptations.

The mature sequences of XcAP-1, XcAP-2, and XcAP-3 exhibited notably low similarity when compared to the human prostatic acid phosphatase (PDB: 1ND6), with values ranging from 17.9% to 22.8%. In contrast, their similarities to each other were higher, ranging from 72.6% to 76.7%. Remarkably, critical catalytic residues, such as His[12] (corresponding to the numbering of 1ND6 sequence) situated within the RHG motif, were found to be replaced by a Gly residue in XcAP-1, −2, and −3. Furthermore, His[257] was replaced by a Ser residue in XcAP-1 and a Pro residue in XcAP-2 and XcAP-3 (Fig. 1a). The absence of both essential catalytic residues strongly suggests the absence of expected catalytic activity in *X. cheopis* salivary acid phosphatases. To verify this hypothesis, we proceeded to obtain the recombinant forms of XcAP-1, −2, and −3 utilizing the HEK293 and Expi293 cell lines. The recombinant proteins were subjected to purification via two chromatographic steps, and their identity was confirmed through N-terminal sequencing (Fig. 1a). Upon examination of the purified recombinant proteins using NuPAGE, a single protein band near the theoretical molecular weight for XcAP-1 (40.3 kDa), XcAP-2 (40.7 kDa) and XcAP-3 (41.1 kDa) was evident (Fig. 1b).

**XcAP-1, −2, and −3 lack the catalytic activity associated with the acid phosphatase family.** It should be noted that the substitution of a catalytic residue does not necessarily result in the loss of catalytic activity. In some cases, these mutations can lead to the emergence of a novel catalytic activity or be accommodated by the enzyme, allowing another residue to serve as the catalytic one. An interesting example can be found within the histidine phosphatase superfamily. Substitution of the catalytic histidine residues of the rat fructose-2–6-biphosphatase for an

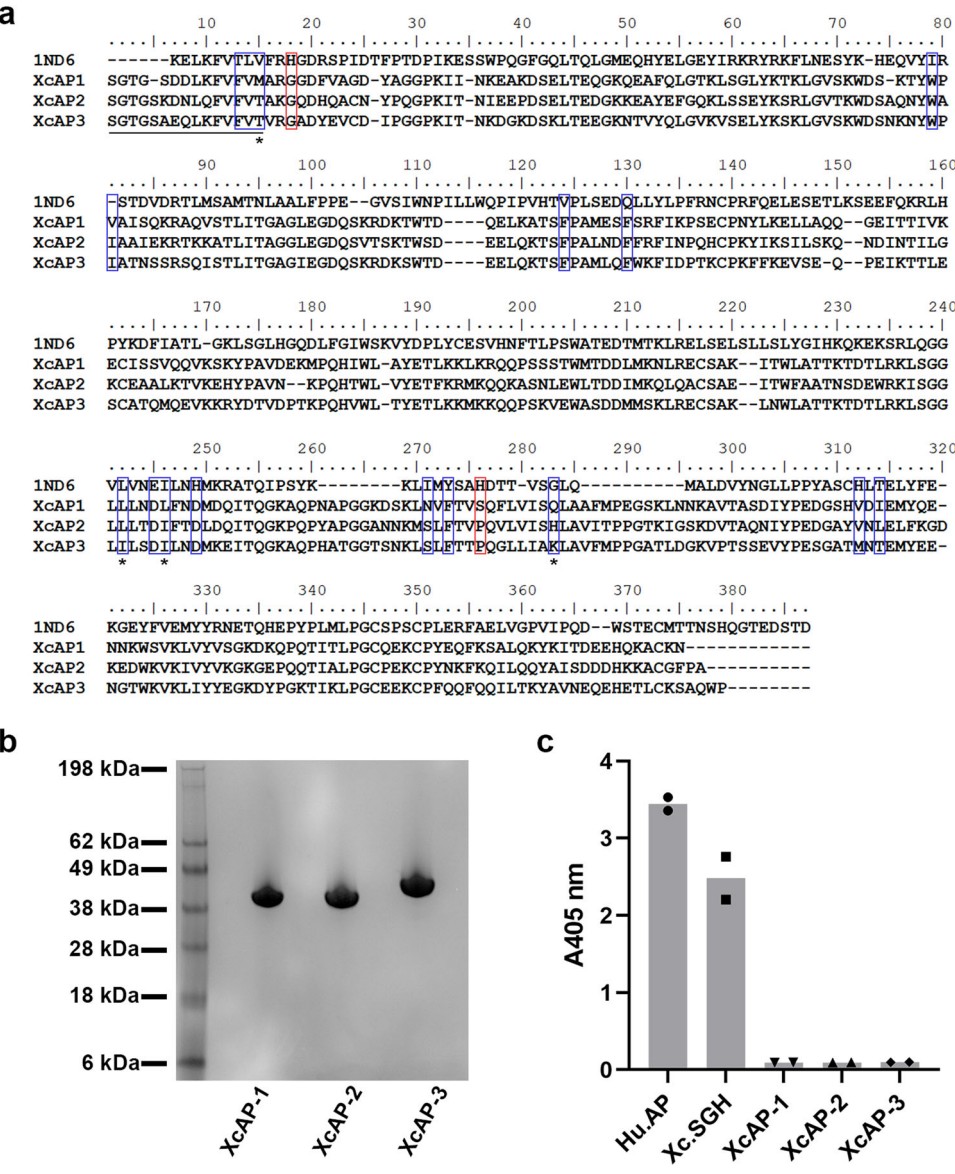

**Fig. 1 X. cheopis pseudo-acid phosphatase primary structure. a** Amino acid alignment of *X. cheopis* recombinant acid phosphatases with the human prostatic acid phosphatase (PDB: 1ND6). The residues identified by N-terminal sequencing are underscored. The putative catalytic residues are highlighted in red and the residues that forms the biogenic amine binding pocket are shown in blue. **b** NuPAGE (4 – 12%) of recombinant XcAP-1, −2 and −3. **c** Measurement of the acid phosphatase activity using pNPP (para-nitrophenyl phosphate) as substrate with *X. cheopis* salivary gland homogenates (Xc.SGH: 0.1 mg/ml) and the recombinant XcAP-1, −2, and −3 (1 µM). The human prostatic acid phosphatase (Hu.AP) was used as positive control. The experiment was conducted in technical duplicates.

alanine decreased the enzymatic activity by less than a factor of 10[28]. Another example can be observed in the cattle tick *Rhipicephalus microplus*, where an aspartic peptidase lacking one of the catalytic aspartic acid residues still maintains its proteolytic activity towards hemoglobin[29]. Together these data reinforce the necessity of confirming the absence of the expected enzymatic activity when working with putative enzymes lacking the canonical catalytic residues.

In our study, we examined the purified recombinant acid phosphatases as well as *X. cheopis* salivary gland homogenates (SGH) ability to cleave p-Nitrophenylphosphate (pNPP). The results showed that XcAP-1, −2, and −3 were unable to process pNPP (Fig. 1c and Fig. S2B–D), confirming that these acid phosphatase-like proteins do not possess the expected catalytic activity typically found in other acid phosphatases. Further investigation of the phosphatase activity in *X. cheopis* SGH

indicated an optimal pH range between 6 and 7 (Fig. S2A). Notably, this activity was strongly inhibited by the acid phosphatase inhibitor, sodium fluoride. Moreover, it was partially inhibited in the presence of sodium tartrate (an acid phosphatase inhibitor) and p-bromolevamisole oxalate (an alkaline phosphatase inhibitor) (Fig. S2E). These findings collectively suggest the presence of active acid phosphatases in *X. cheopis* SGH, with some of them exhibiting resistance to sodium tartrate. In our previous studies of *X. cheopis* salivary gland homogenates, we also identified transcripts containing the RHG motif but lacking the second histidine residue (File S1). Since some proteases lacking one of their catalytic residues have been shown to maintain their enzymatic activity[29,30], it is possible that the observed activity in *X. cheopis* SGH (Fig. 1c) is due to such proteins.

Functionally, acid phosphatases and their catalytic activities have been demonstrated to be present in the salivary glands of other

hematophagous arthropods. Nevertheless, their specific role in the blood-feeding process remains largely unclear. Acid phosphatases have been reported in hard ticks, such as *R. microplus*[31] and *Hyalomma anatolicum anatolicum*[32]. In the case of ticks, it was observed that the concentration of this enzyme increases in the salivary glands as feeding progresses and authors have speculated that such enzymes may be linked to the degenerative process of tick salivary glands through apoptosis. In kissing bugs, acid phosphatase activity has been detected in the salivary glands of both male and female *Triatoma infestans*, *Panstrongylus megistus* and *Rhodnius neglectus*[33]. Notably, acid phosphatase activity was found in both the nucleus and cytoplasm of salivary gland cells, suggesting that these enzymes may serve multiple roles in the physiology of kissing bugs. However, further biochemical and functional characterization of such enzymes is necessary to provide a deeper insight into their precise function in the context of blood acquisition.

**XcAP-1, XcAP-2, and XcAP-3 are kratagonists.** Previously, it was suggested that the absence of the expected catalytic activity in salivary acid phosphatases would result in the protein being permanently bound to its natural substrate, thereby limiting its availability[25]. This chelating activity has been characterized in various hematophagous vectors, including mosquitoes, ticks, triatomines, sand flies, and horseflies. Salivary proteins in these vectors have the ability to bind with high affinities to agonists that are important for host hemostasis and inflammatory response, such as biogenic amines and eicosanoids[4,34–37].

To counteract the hemostatic response induced by these agonists, it has been speculated that the chelating protein must reach concentrations between 0.2 and 2 µM in the feeding site, which corresponds to the normal receptor saturating concentration of histamine and serotonin[2]. Considering an average molecular weight of 40 kDa for salivary acid phosphatases, this concentration range translates to 8–80 ng/µl. It was estimated that a pair of *X. cheopis* salivary glands contains ~1.5 µg of total protein, with acid phosphatases accounting for ~40% of the proteins present in flea salivary gland homogenates[18,25]. This corresponds to a total of 0.6 µg of acid phosphatases. Assuming that the average volume of the bite site is equivalent to the average volume of ingested blood, which has been determined to be 0.12 µl for *X. cheopis*[38], we can estimate that the concentration of acid phosphatases would be up to 5 µg/µl. Even if only a small fraction of the salivary gland content is secreted during a feeding cycle, it is still possible to achieve concentrations above the necessary range for effective chelation.

The ability of XcAP-1, −2, and −3 to bind small agonists was evaluated using isothermal titration calorimetry (ITC). The results demonstrated a high affinity of these proteins towards both biogenic amines and leukotrienes, as summarized in Table 1. Importantly, all the assays consistently revealed stoichiometries ranging from 0.7 to 1.3, indicating the presence of a single binding site for each ligand tested (Figs. S3–S5). XcAP-1 demonstrated the ability to interact with various biogenic amines, while XcAP-2 and XcAP-3 specifically bound to histamine. Moreover, distinct affinities were observed for leukotrienes. Recombinant XcAP-2 exhibited comparable affinities for LTB$_4$, LTC$_4$, LTD$_4$, and LTE$_4$, whereas XcAP-1 and −3 selectively bound to LTC$_4$.

From the host perspective, biogenic amines and leukotrienes play an important role in the hemostatic and inflammatory responses. For instance, serotonin is involved in platelet aggregation[39] and vasoconstriction[40], while histamine is a potent inducer of pain[41], potentially alerting the host to the presence of the vector. The injection of LTD$_4$ and LTE$_4$ into the human skin has been shown to induce erythema formation[42], while LTB$_4$ acts

**Table 1 Dissociation constants (nM) of *X. cheopis* acid phosphatases with different ligands determined by isothermal titration calorimetry (ITC) assays.**

|  | XcAP-1 | XcAP-2 | XcAP-3 | Human PAP[a] |
|---|---|---|---|---|
| Serotonin | 3.72 | - | - | - |
| Norepinephrine | 76.9 | - | - | - |
| Epinephrine | 17.7 | - | - | - |
| Histamine | - | 37.7 | 37.0 | - |
| LTB$_4$ | - | 110.8 | - | - |
| LTC$_4$ | 37.4 | 78.7 | 65.8 | - |
| LTD$_4$ | - | 152.9 | - | - |
| LTE$_4$ | - | 60.2 | - | - |
| U-46619 | - | - | - | N.T |
| cTXA$_2$ | - | - | - | N.T |
| ADP | - | - | - | N.T |

Dashes (-) indicated that no binding was observed.
*N.T.* Not tested.
[a]Human prostatic acid phosphatase.

as a potent chemoattractant for leukocytes[43]. Moreover, leukotriene signaling can lead to plasma leakage at the bite site, resulting in the dilution of the blood meal rich in red blood cells[44]. However, in the presence of sufficient concentrations of XcAP-1, −2 and −3, these host responses could be partially or completely blocked, thereby facilitating successful blood acquisition by the vector. Finally, the chelation of LTC$_4$ by different kratagonists could be an efficient way to block the overall leukotriene production and downstream signaling as LTC$_4$ is enzymatically converted to LTD$_4$ and LTE$_4$[45].

Since XcAP-1, −2 and −3 can bind to biogenic amines and leukotrienes we decided to investigate whether these ligands share the same binding site. To test this hypothesis, we conducted an ITC competition assay measuring the biogenic amine binding to XcAP-1, −2, and −3, which had been pre-incubated with saturating concentrations of LTC$_4$ (Fig. 2). In all assays the thermodynamic parameters obtained from the competition assay were almost identical to those observed when LTC$_4$ was absent (Table 2). These results indicate that XcAP-1, −2, and −3 can independently bind both biogenic amines and leukotrienes, indicating the presence of distinct binding sites for each ligand. Lastly, we investigated the binding of agonists to the catalytic active human prostatic acid. Notably, no interaction was observed in all cases tested (Table 1), indicating that the chelating activity exhibited by XcAP-1, −2, and −3 is not reminiscent of the histidine acid phosphatase superfamily. Instead, it suggests that these flea salivary acid phosphatase-like proteins have undergone a distinct evolutionary process, wherein they have been adapted to function as kratagonists.

**XcAP-1 possesses a specific pocket that is capable of binding biogenic amines.** The three-dimensional structure model of XcAP-1 complexed with serotonin was determined using X-ray crystallography data at a resolution of 1.75 Å (Table 3). The crystal belonged to the monoclinic space group P2$_1$, and the asymmetric unit contained two near-identical XcAP-1:serotonin complexes (r.m.s.d = 0.230 Å over 356 Cα positions, Fig. S6E). In both complexes we observed high-quality electron density spanning from XcAP-1 residues Asp[7] (numbering based on the mature sequence) to Asn[361] and including the serotonin ligand (Fig. S6C). Interestingly, additional interpretable electron density near the putative catalytic site of XcAP-1 was also observed, in which we modeled a palmitoleic acid molecule (Fig. S6D). It is noteworthy that palmitoleic acid was not included in any buffer during the purification and crystallization preparations, therefore,

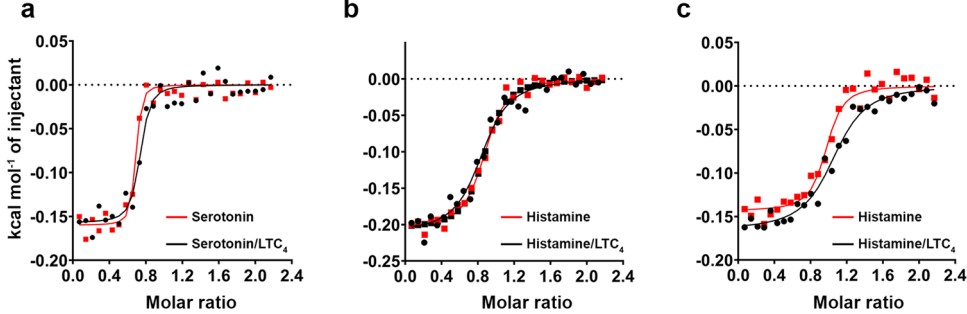

**Fig. 2 Isothermal titration calorimetry competition assay.** Measurement of biogenic amine (15 µM) binding to (**a**) XcAP-1 (1.5 µM), (**b**) XcAP-2 (1.5 µM) and (**c**) XcAP-3 (1.5 µM) in the presence (black dots) or absence (red dots) of LTC$_4$ (6 µM). The data points (dots) represent the injection enthalpies observed during the assay. A single-site binding model was fitted to the data (lines).

**Table 2 Thermodynamic parameters of XcAP-1, −2, and −3 binding to biogenic amines in the presence or absence of saturating concentrations of LTC$_4$.**

| Protein | Ligand | $K$ (M$^{-1}$) | $\Delta H$ (cal/mol) | $\Delta S$ (cal/mol/deg) |
|---|---|---|---|---|
| XcAP-1 | Serotonin | $1.69 \times 10^8 \pm 8.97 \times 10^7$ | $-1.573 \times 10^4 \pm 536.1$ | − 14.2 |
| | LTC$_4$/Serotonin | $5.48 \times 10^8 \pm 3.01 \times 10^7$ | $-1.598 \times 10^4 \pm 390.6$ | − 12.7 |
| XcAP-2 | Histamine | $2.28 \times 10^7 \pm 6.08 \times 10^6$ | $-2.112 \times 10^4 \pm 718.5$ | −36.0 |
| | LTC$_4$/Histamine | $4.07 \times 10^7 \pm 7.83 \times 10^6$ | $-2.057 \times 10^4 \pm 397.2$ | −33.0 |
| XcAP-3 | Histamine | $3.43 \times 10^7 \pm 9.18 \times 10^6$ | $-1.509 \times 10^4 \pm 402.3$ | −15.3 |
| | LTC$_4$/Histamine | $7.35 \times 10^7 \pm 2.86 \times 10^7$ | $-1.436 \times 10^4 \pm 434.2$ | −11.4 |

**Table 3 Data collection, phasing, and refinement statistics for XcAP-1 complexed with serotonin.**

| | XcAP-1:5HT |
|---|---|
| **Data collection** | |
| Space group | P2$_1$ |
| Cell dimensions | |
| $a, b, c$ (Å) | 47.97, 71.90, 113.51 |
| $\alpha, \beta, \gamma$ (°) | 90.00, 91.54, 90.00 |
| Resolution (Å) | 44.54–1.75 |
| $R_{merge}$ | 14.0 (66.3) |
| $I / \sigma I$ | 4.9 (1.11) |
| Completeness (%) | 0.97 (0.60) |
| Redundancy | 5.1 (4.0) |
| **Refinement** | |
| Resolution (Å) | 44.54–1.75 |
| No. reflections | 375,185 |
| $R_{work}/R_{free}$ | 0.2154/0.2503 |
| No. atoms | |
| Protein | 5516 |
| Ligand/ion | 100 |
| Water | 300 |
| $B$-factors | |
| Protein | 20.94 |
| Ligand/ion | 22.64 |
| Water | 22.19 |
| R.m.s. deviations | |
| Bond lengths (Å) | 0.007 |
| Bond angles (°) | 0.903 |

it is likely that it originated from the HEK293 cell culture used to obtain the recombinant protein.

Similar to the rat acid phosphatase[46] (PDB:1RPA) and the human prostatic acid phosphatase[47] (PDB: 1ND6), each XcAP-1 complex consists of two domains: the larger α/β domain, which is comprised of β-sheets and α-helices, and the smaller α-domain, which consists of α-helices and loops (Fig. S6E). When compared to the monomeric structure of human prostatic acid phosphatase we obtained a r.m.s.d of 1.948 Å (against chain A, over 354 Cα positions), indicating an overall conserved three-dimensional structure despite the low identity between the two sequences (Fig. S6B). However, one notable difference between XcAP-1 and the human prostatic acid phosphatase is the absence of β-strand 3 (Fig. S6B), which is known to be involved in the homodimer formation of the rat enzyme[46] and is essential for its catalytic activity[48]. This suggests that the loss of β-strand 3 in XcAP-1 may represent an evolutionary adaptation that enables the protein to function as a monomeric kratagonist.

In the current structure model, the serotonin molecule was found within a pocket located in the α/β-domain (Fig. S6E). It is worth noting that this pocket is absent in the human prostatic acid phosphatase structure, explaining its inability to interact with biogenic amines (Table 1). The entry to the binding pocket is formed by the side chains of XcAP-1 residues Phe[111], Asp[227] and Asp[231]. Notably, both Asp[227] and Asp[231] form salt bridges with the protonated aliphatic amino group of serotonin, contributing to its stabilization (Fig. 3a). Additionally, a water molecule located within the binding site entry forms a hydrogen bond between the amino group of serotonin and the side chain of Asn[253], further trapping the biogenic amine at the pocket. Within the pocket, the serotonin indole group is surrounded by hydrophobic and aromatic residues, creating a hydrophobic environment. At the bottom of the pocket, the serotonin hydroxyl group forms hydrogen bonds with both Val[8] and Asn[253], while the indole nitrogen interacts with the side chain of Gln[256] (Fig. 3b). Interestingly, despite the fact that XcAP-1 and the D7 protein in mosquitoes belong to different protein families, the pocket structure and the observed interactions in the XcAP-1:serotonin complex bear similarities to those described in the D7 proteins[49,50].

During the comparison of the primary structures of XcAP-1, −2, and −3, it was observed that the residues forming the bottom of the serotonin binding pocket showed a lack of conservation (Fig. 1a), which likely contributes to their different affinities for biogenic amines (Table 1). Since crystals were not obtained for XcAP-2 and

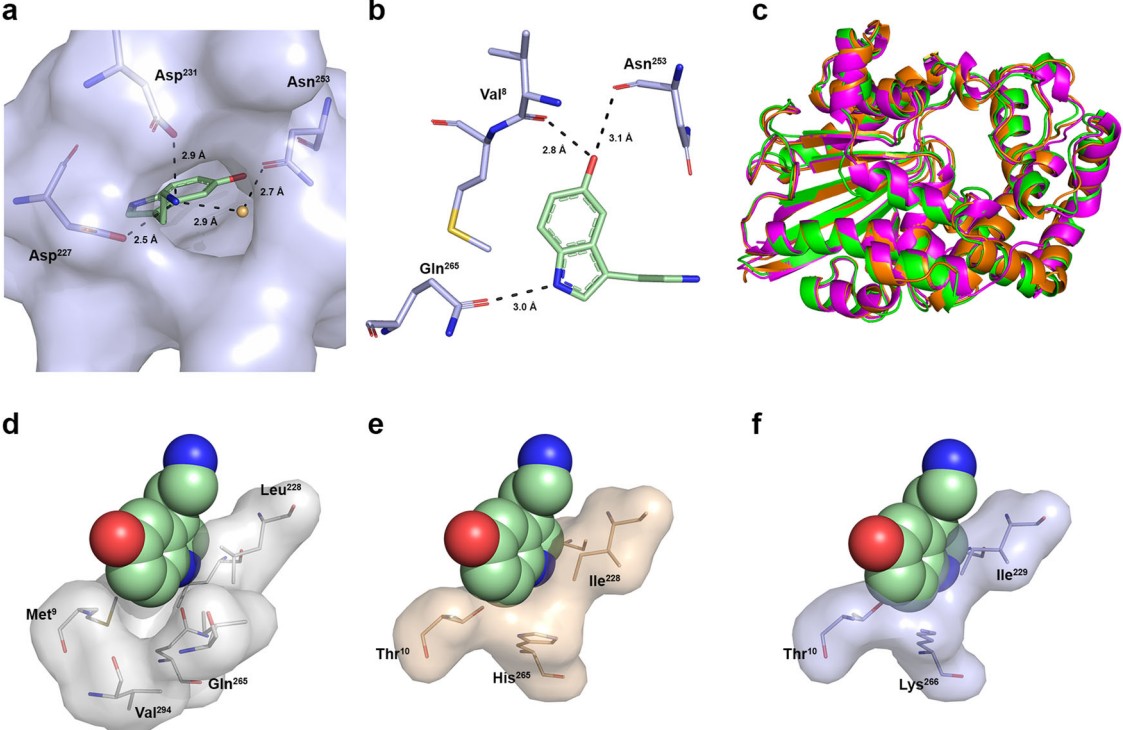

**Fig. 3 Characterization of the XcAP-1 serotonin binding site. a** Interactions between XcAP-1 and serotonin at the entry of the binding pocket. Water molecule is represented as a golden sphere. **b** Interactions between serotonin and the residues at the closed end of the binding pocket. **c** Superposition of the crystal structure of XcAP-1 (green) with the AlphaFold2 models of XcAP-2 (orange) and XcAP-3 (magenta). Comparison of (**d**) XcAP-1, (**e**) XcAP-2 and (**f**) XcAP-3 residues that form the bottom of the biogenic amine binding site. The fixed position of serotonin (spheres) was based on that observed in the crystal complex. Nitrogen atoms are represented in blue, oxygen atoms in red and sulfur atoms in yellow.

−3, three-dimensional structure models were generated using Alphafold2 and superimposed onto the crystal structure of the XcAP-1 complex. Overall, the structures of XcAP-2 and −3 closely resembled that of XcAP-1 (Fig. 3c). However, upon closer examination of the serotonin binding site, it was evident that XcAP-2 and −3 cannot accommodate the biogenic amine in the same manner as XcAP-1. In XcAP-1, the residues Met[9], Leu[228], and Gln[265] show good shape complementary with the serotonin indole group, and the pocket is closed by the Val[294] residue, providing support for the serotonin molecule (Fig. 3d). In XcAP-2, these residues are substituted with Thr[10], Ile[228], and His[265], which have side chains that clash with the serotonin indole group in its XcAP-1 binding mode (Fig. 3e). A similar conflicting structure was observed in the XcAP-3 Alphafold2 model. In addition to the Met[9]/Thr[10] and Leu[228]/Ile[229] substitutions observed in XcAP-2, the side chain of Lys[266] (corresponding to Gln[265] in XcAP-1) extends towards the aromatic ring of serotonin, resulting in a shallower bottom of the binding pocket (Fig. 3f). These substitutions at the bottom of the binding pocket effectively reduce its volume, suggesting that XcAP-2 and −3 may only accommodate smaller biogenic amines, such as histamine, in their binding pockets.

**XcAP-1 "catalytic site" can bind to fatty acids.** Surprisingly, in addition to the serotonin ligand, interpretable density for a second ligand was found near the mutated catalytic residues. The tubular shape of this density bore resemblance to that observed in the crystal structure of tablysin-15, a salivary member of the cysteine-rich/antigen 5/pathogenesis-related 1 (CAP) protein family from the blood feeding horse fly *Tabanos yao*[4]. The unknown ligand was identified as 16-carbon palmitic acid, that apparently became bound during the preparation of the recombinant protein. Furthermore, the same study demonstrated that tablysin-15 can

scavenge leukotrienes, and its crystal structure complexed with LTE$_4$ revealed similar interactions between the 16-carbon palmitic acid and the hydrocarbon chain of LTE$_4$. Based on these findings, it is plausible to propose that the region near XcAP-1's substituted catalytic residues, where the fatty acid is bound, serves as the binding site for leukotrienes.

In the XcAP-1 ligand density, a 16-carbon fatty acid could be modeled. The carboxyl group of the fatty acid is well defined and is oriented towards the substituted catalytic residues, while the hydrocarbon chain extends into the interior of the pocket. An approximately 90-degree bend occurs at C9 which is consistent with the presence of a *cis* unsaturation at this position (Fig. 4a). We therefore modeled the ligand as palmitoleic acid, a 16-carbon fatty acid containing a *cis* unsaturation at C9. The fatty acid's binding mode is predominantly shaped by the side chains of hydrophobic residues, including Ala[21], Gly[22], Phe[120], Ile[121], Leu[130], Leu[133], Pro[166], Trp[170], Leu[171], Ile[205], Ala[209], Phe[260] and Ile[263] (Fig. 4a). However, an extended solvent-accessible area can be observed in the direction of XcAP-1's substituted catalytic residues, partially occupied by a SO$_4$ molecule (Fig. 4b).

Upon comparing the XcAP-2 and −3 AlphaFold models to XcAP-1, noticeable differences in the overall structure of the putative fatty acid binding pocket are apparent (Fig. 4c, d). Notably, XcAP-2's pocket does not appear to be as elongated as the one observed in XcAP-1, additionally, the side chains of Ile[132], Leu[136], Ile[135], Leu[142], Pro[166] and Gln[167] are positioned in such a way to create additional space near the fatty acid 90° bend (Fig. 4c) compared to XcAP-1. It is possible that this wider pocket might better accommodate the eicosanoid molecule, partially explaining the different affinities observed in our ITC assays (Table 1). Furthermore, XcAP-3's fatty acid pocket appears to be quite narrow, with the side chains of Ile[123], Phe[131], and Phe[132] slightly touching

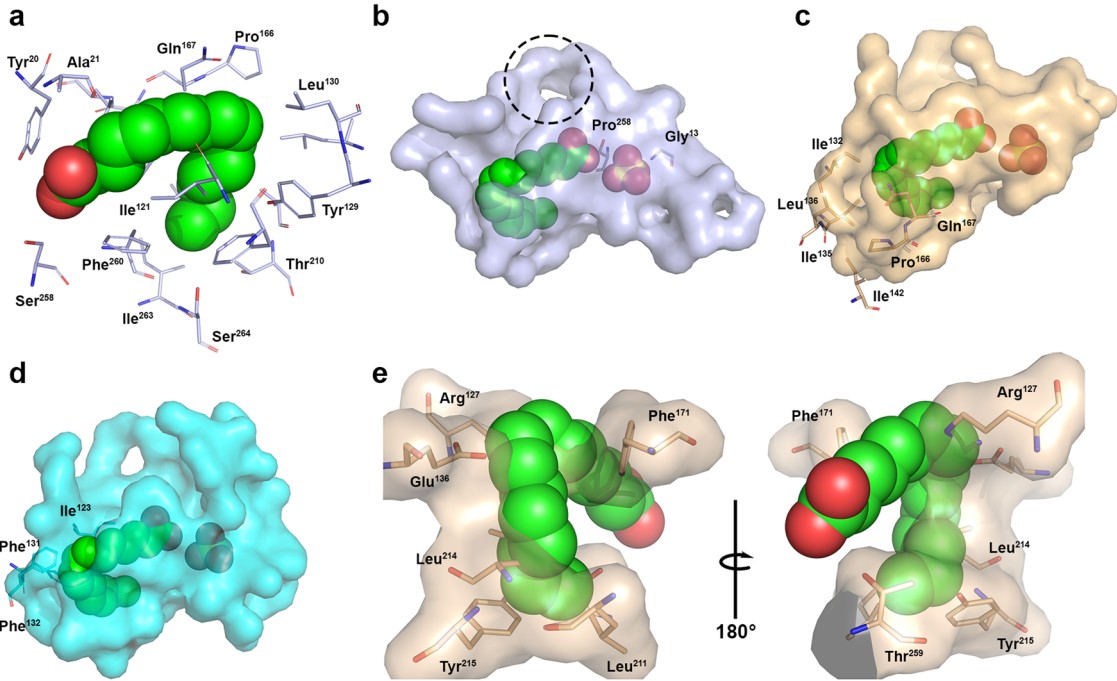

**Fig. 4 Characterization of the fatty acid binding site in XcAP-1. a** Hydrophobic binding site residues of XcAP-1 depicted as light blue sticks, with the palmitoleic acid molecule shown as green spheres. **b** Surface representation of the putative leukotriene binding pocket in XcAP-1. Substituted catalytic residues shown in sticks, and the $SO_4$ atom represented as an orange sphere. The dotted circle highlights the solvent-accessible cleft near XcAP-1's Lys[31] and Arg[78] residues. Surface representation of the putative leukotriene binding pocket of (**c**) XcAP-2 and (**d**) XcAP-3 based on the superposition of the AlphaFold model onto the XcAP-1 crystal model. **e** Superposition of the human prostatic acid phosphatase (PDB: 1ND6) onto the XcAP-1 crystal model, highlighting the clashes between the side chain of the human enzyme and the 16-carbon palmitic acid in its XcAP-1 binding mode. Nitrogen atoms represented in blue, and oxygen atoms in red.

the 16-carbon palmitic acid in its XcAP-1 binding mode (Fig. 4d). It's likely that small structural modification take place upon leukotriene binding to accommodate the eicosanoid molecule.

Finally, the superimposition of the human prostatic acid phosphatase structure (PDB: 1ND6) onto the XcAP-1 model provides an insight into the molecular basis of the catalytic active enzyme's inability to bind eicosanoids. Notably, in the human enzyme the side chains of Arg[127], Glu[136], Phe[171], Leu[211], Leu[214], Try[215], and Thr[259] create significant clashes with palmitoleic acid molecule situated as in XcAP-1, particularly near the 90° bend of the fatty acid in its XcAP-1 binding mode (Fig. 4e). Consequently, it appears that these clashes render the human enzyme structurally incompatible with leukotriene binding.

It is important to mention that LTC₄, LTD₄, and LTE₄ share the same 20-carbon hydrocarbon chain but differ in the peptide group bound at C6 via a thioether linkage. Assuming that the eicosanoid interacts similarly to the 16-carbon palmitoleic acid, it is conceivable that the additional carbons from the leukotriene molecule extend in the direction of the substituted catalytic residues. In this scenario, the region currently occupied by the palmitic acid carboxyl group would be filled by the leukotriene C5 or C6, similar to the tablysin-15-LTE₄ crystal structure[4]. Furthermore, the peptide substituent may extend along the solvent-accessible cleft formed by XcAP-1's Lys[31] and Arg[78] side chains, situated at the junction of the α/β and α domains (Fig. 4b).

Currently, it is challenging to fully account for the contributions of the substituted catalytic sites to eicosanoid binding, as our attempts to obtain a complexed crystal and reconstituted catalytically active XcAP-1, −2, and −3 were unsuccessful. Nevertheless, despite these limitations, it is evident that *X. cheopis* salivary acid phosphatase-like proteins exhibit additional structural alterations that are not present in catalytic members of the acid phosphatase

family. Though we are unable to directly observe the specific interactions between the eicosanoid and XcAP-1, the presence of the structurally distinct yet analogous hydrocarbon chain near the substituted catalytic residues strongly suggests the likelihood of the leukotriene molecule interacting within this region. The overall alterations observed in XcAP-1, −2, and −3 collectively form a unique binding pocket that can accommodate the hydrocarbon chain of eicosanoid molecule.

## Conclusion

In the present study, we have conducted a comprehensive characterization of acid phosphatase-like proteins found in the salivary glands of the flea, *X. cheopis*. Upon analyzing their primary structure, we classified XcAP-1, −2, and −3 as potential members of the histidine phosphatase superfamily. However, the three flea proteins exhibit substitutions in the catalytic residues that are typically conserved among active members of the histidine phosphatase superfamily. Our data revealed that these flea salivary acid phosphatase-like proteins have lost their expected catalytic activity and, instead, have adopted a role as scavengers of biogenic amines and leukotrienes. Building on this newfound functionality, we propose that XcAP-1, −2, and −3 may contribute to the acquisition of blood by sequestering agonists relevant to the host's hemostatic and immune response (Fig. 5).

## Materials and methods

**Expression and purification of recombinant acid phosphatases.** The open reading frame of mature XcAP-1 was cloned into the VR1020 vector, while the open reading frames of mature XcAP-2 and XcAP-3 were cloned into the pcDNA3.1(+) vector (GenScript). The recombinant acid phosphatases were produced in the HEK293

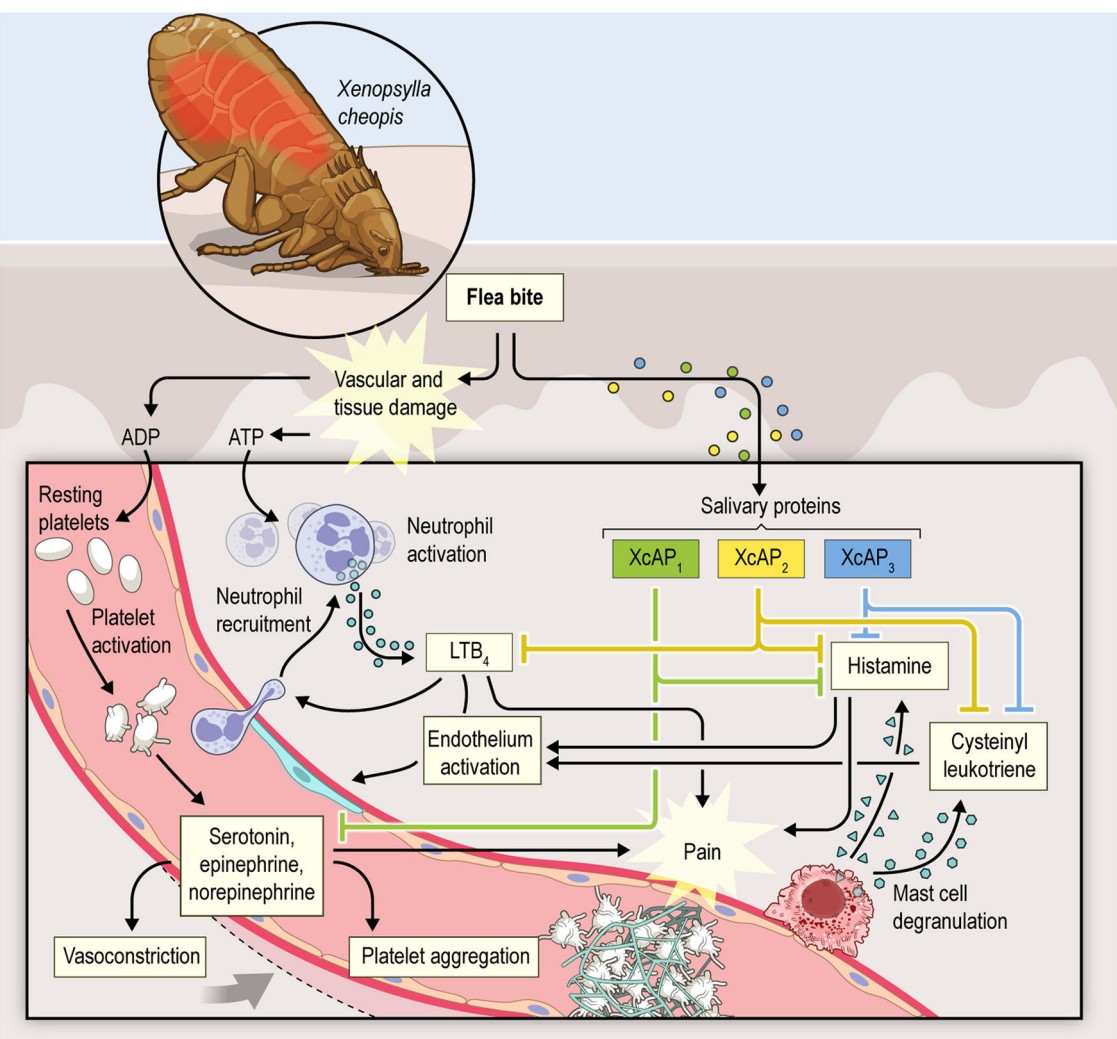

**Fig. 5 Proposed mechanism of action of *X. cheopis* salivary acid phosphatase-like proteins.** Schematic illustrating the proposed mechanism of action for XcAP-1, −2, and −3 in modulating host hemostatic and immune response at the bite site.

cell line and protein purification was performed using an ÄKTA Pure system (GE) as follows: The cell culture supernatant was concentrated and dialyzed in 25 mM sodium acetate buffer pH 5.6 overnight and applied into a Hiprep SP resin equilibrated with the same buffer. Protein elution was performed by a 0–100% linear gradient of 25 mM sodium acetate pH 5.6 containing 1.0 M NaCl. Fractions containing the protein of interested were pooled, dialyzed in 50 mM phosphate buffer pH 6.0 containing 1.5 M ammonium sulfate, and applied to a Phenyl FF resin. Protein elution was carried out with a linear gradient 0–100% of 50 mM phosphate buffer pH 6.0. Purified recombinant acid phosphatases were inspected following electrophoresis in a NuPAGE gel (4–12%), and the total protein concentration was determined using a BCA kit following the manufacturer's instructions (PIERCE, US). The identity of the purified recombinant acid phosphatases was confirmed by N-terminal sequencing using the Edman degradation method on a Procise 494 sequencer (Applied Biosystems, US) equipped with a PTH-amino acid analyzer. The purified recombinant proteins (1 mg/ml) were deposited on a 45 mm PVDF membrane disk, pre-wetted with methanol, and kept at room temperature until dried. The PVDF membrane was then vortexed twice in 0.1% TFA in water and dried at room temperature. Samples were then loaded into the cartridge block and subjected to Edman degradation. The data were analyzed using the built-in SequencePro software.

**Acid phosphatase catalytic activity measurement**. The catalytic activity of recombinant XcAP-1, −2, and −3, and *X. cheopis* salivary glands homogenates (SGH) was inspected using an acid phosphatase colorimetric assay kit (Abcam, US), following the manufacturer's instructions. Briefly, the substrate pNPP (para-Nitrophenyl phosphate) was freshly prepared in the provided assay buffer and added to the purified recombinant acid phosphatases (1 μM), *X. cheopis* SGH (0.1 mg/ml) and to the supplied human prostatic acid phosphatase solution (10 μl). After an incubation period of 1 h at 30 °C, 20 μl of the supplied stop solution was added to all wells and the absorbance (405 nm wavelength) measured using a VersaMax microplate reader (Molecular Devices). All measurements were performed in technical duplicates and the average O.D of the negative control reaction, which contained only the assay buffer, pNPP and stop solution, was subtracted from all measurements. Finally, the average $A_{405}$ and the standard deviation of the mean of each sample was plotted. Determination of the optimum pH was carried out in 75 mM Tris, 25 mM glycine, 25 mM MES and 25 mM acetic acid with pH varying from 3 to 9 with purified recombinant XcAP-1, −2, and −3 (1 μM) or *X. cheopis* SGH (0.01 mg/ml). Inhibitory activity of *X. cheopis* SGH (0.01 mg/ml) phosphatase activity was performed in 75 mM Tris, 25 mM glycine, 25 mM MES, 25 mM acetic acid pH 7.0 in the presence

of sodium fluoride (100 μM), sodium tartrate (100 μM) or p-Bromolevamisole oxalate (100 μM).

**Ligand-protein screening by isothermal titration calorimetry (ITC).** Purified recombinant proteins and ligands were diluted in PBS buffer pH 7.4 and degassed for 5 min prior to the ITC assays. For the eicosanoid ligands, an aliquot was dried under a constant flow of nitrogen, and then PBS pH 7.4 was added. The sample was briefly vortexed, sonicated for 10 min and degassed for 5 min. The binding experiments were performed on a MicroCal VP-ITC calorimeter (Malvern, UK) at 30 °C. The purified recombinant acid phosphatase was added to the ITC cell to a final concentration of 1.5 μM (assays with biogenic amines) or 3 μM (assays with eicosanoids), while the syringe was filled with the ligand solution (15 or 30 μM). Heats were measured using a 10 μl ligand injection over periods of 20 s with 300 s spacing time between the injections with a total of 28 injections. The competition assays using biogenic amines and $LTC_4$ were carried out by filling the calorimeter cell with the recombinant acid phosphate in addition to $LTC_4$ in a fourfold molar excess, while the syringe was filled with the serotonin, for XcAP-1, or histamine, for XcAP-2 and −3, in addition to the same amount of $LTC_4$ used in the cell. The remaining conditions of the experiments were the same as described above. Data analysis was performed by integrating the heats measured and plotting against the macromolecule molar ratio. A single binding model was fitted to the data and the thermodynamic parameters (equilibrium association constant, enthalpy, and entropy change) were estimated using the MicroCal Origin software.

**X-ray diffraction, data collection, and structure solution.** Purified recombinant XcAP-1 (19.8 mg/ml) was incubated with serotonin (Sigma, US) at 1:1.5 molar ratio (XcAP-1:Serotonin) at 37 °C for 30 min. The complex was crystalized using the hanging drop vapor diffusion method in 0.1 M MES pH 5.4, 3.0 M $(NH_4)_2SO_4$ at room temperature. Mature crystals were obtained after a 7-day incubation period and were flash cooled in liquid nitrogen in 0.1 M MES pH 5.4, 3.7 M $(NH_4)_2SO_4$. Diffraction data were collected with the Southeast Regional Collaborative Access Team (SER CAT, Beamline 22-ID, Wavelength 1.000 Å) at the Advanced Photon Source (Argonne National Laboratory) and processed using HKL2000[51]. The complex crystallized in the space group $P2_1$ with two complexes contained in the asymmetric unit (Table 2). The structure complex was solved by molecular replacement with Phaser from the CCP4i suit (8.0.006) using the XcAP-1 model structure predicted by AlphaFold2[52]. The model was manually corrected using Coot (0.9.8.1)[53] and refined with the phenix.refine (1.20.1)[54] tool with a TLS model. Alphafold2 was also used to obtain tridimensional models of mature XcAP-2 and XcAP-3.

**Phylogenetic analysis.** The phylogenetic tree was constructed using the maximum likelihood method and JTT matrix-based model[55] with a bootstrap consensus from 500 replicated using MEGA11[56].

**Reporting summary.** Further information on research design is available in the Nature Portfolio Reporting Summary linked to this article.

## Data availability

Coordinates and structure factors for the XcAP-1:serotonin complex have been deposited in the wwPDB under the accession number 8GDL. The source data utilized for Fig. 1c and Fig. 2 are also available as supplementary data 2.

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

## Acknowledgements

We would like to acknowledge Ryan Kissinger from the Visual and Medical Arts Branch (RTB, NIAID, NIH) for his work on the illustration describing the proposed mechanism of action of X. cheopis salivary acid phosphatase-like proteins. This work was supported by the Intramural Research Program of the Division of Intramural Research, National Institute of Allergy and Infectious Diseases (NIAID), National Institutes of Health (NIH), and the staff at SER CAT, Advanced Photon Source, Argonne National Laboratory for assistance with data collection. This work utilized the computational resources of the NIH HPC Biowulf cluster (http://hpc.nih.gov). This work was supported by the intramural research program of the NIAID, National Institutes of Health. The content is solely the responsibility of the authors and does not necessarily represent the official views of the National Institutes of Health.

## Author contributions

Experiments were designed by S.L., J.F.A., B.J.H., and J.M.R.; carried out by S.L., C.F.B., and J.F.A.; and analyzed by S.L., J.F.A., B.J.H., and J.M.R.; S.L. wrote the paper, and S.L., J.F.A., C.F.B., B.J.H., and J.M.R. revised the paper.

## Funding

## Competing interests

The authors declare no competing interests.
