## [Peer Review File · Communications Biology]

Reviewers' comments:

Reviewer #1 (Remarks to the Author):

The work by Lu and others describes the functional and structural characterization of acid phosphatases-like proteins from the flea *Xenopsylla cheopis*. Three flea salivary acid phosphatases-like proteins, XcAP-1, -2, and -3, lost their expected catalytic activity and became scavengers of biogenic amines and leukotrienes. Moreover, X-ray crystallography data from XcAP-1 complexed with serotonin revealed insights into their binding mechanisms. The manuscript is well written, the methods section is clearly documented to allow replication studies, and the quality of the figures is satisfactory. My first suggestion is to reduce background description in the abstract section which seems unnecessarily long. Secondly, if the content related to "the cat flea, *Ctenocephalides felis*, salivary acid phosphatases-like transcripts are likely kratagonists" is included in the manuscript, it is inappropriate to emphasize the species, *Xenopsylla cheopis*, in the title. Alternatively, adding *Ctenocephalides felis* in the title, too. In addition, it is necessary to add functional experiments and results related to proteins from the cat flea, *Ctenocephalides felis* in the section if *Ctenocephalides felis* is included in the title. If the present title is used, the contents related to the salivary acid phosphatases-like transcripts of the cat flea should be removed.

Reviewer #2 (Remarks to the Author):

Journal: Nature

Reviewer: Aguirre-Garcia MM Date: 09/23/2023

Title: Acid phosphatases-like proteins, a biogenic amine and leukotriene-binding salivary protein family from the flea *Xenopsylla cheopis*.

In this paper the analysis of flea *X. cheopis* salivary acid phosphatases-like proteins lost their expected catalytic activity and became scavengers of biogenic amines and leukotrienes.

In a novel contribution regarding the presence of acid phosphatases in flea salivary glands, the authors provide a detailed description of the interaction between recombinant phosphatase proteins and bioactive molecules. I believe this manuscript could be accepted for publication with some suggestions:

Comments on Introduction:

1. Elaborate further on the concepts of phosphatases, including their definition and classification.
2. Provide a more comprehensive account of other studies on the presence of phosphatases in arthropod salivary glands.

Comments on Methods:

It is recommended to supplement your study with some biochemical analyses of the acid phosphatase protein, such as:

1. Determine or mention the protein profile present in the salivary glands and the recombinant proteins by performing SDS-PAGE gel electrophoresis, visualizing the proteins through Coomassie blue staining or silver staining.
2. Characterize the phosphatase proteins in salivary gland extracts and recombinant proteins by using specific inhibitors of their activity, which have been well-established. Analyzing the acid phosphatase activity in the presence of a group of phosphatase inhibitors is suggested.
3. Determine the optimal pH of the generated recombinant proteins to establish whether they are indeed acid phosphatases, as described by the authors.

Comments on Conclusions:

Propose a potential model for the interaction of acid phosphatase with these inflammatory mediators.

Reviewer #1 (Remarks to the Author):

The work by Lu and others describes the functional and structural characterization of acid phosphatases-like proteins from the flea *Xenopsylla cheopis*. Three flea salivary acid phosphatases-like proteins, XcAP-1, -2, and -3, lost their expected catalytic activity and became scavengers of biogenic amines and leukotrienes. Moreover, X-ray crystallography data from XcAP-1 complexed with serotonin revealed insights into their binding mechanisms. The manuscript is well written, the methods section is clearly documented to allow replication studies, and the quality of the figures is satisfactory. My first suggestion is to reduce background description in the abstract section which seems unnecessarily long. Secondly, if the content related to “the cat flea, *Ctenocephalides felis*, salivary acid phosphatases-like transcripts are likely kratagonists” is included in the manuscript, it is inappropriate to emphasize the species, *Xenopsylla cheopis*, in the title. Alternatively, adding *Ctenocephalides felis* in the title, too. In addition, it is necessary to add functional experiments and results related to proteins from the cat flea, *Ctenocephalides felis* in the section if *Ctenocephalides felis* is included in the title. If the present title is used, the contents related to the salivary acid phosphatases-like transcripts of the cat flea should be removed.

Reply: We would like to express our gratitude to the reviewer for dedicating their time and expertise to enhance our manuscript. In response to the feedback provided, we have streamlined the background information within the abstract. Additionally, as suggested by the reviewer, we have removed the section pertaining to cat flea salivary acid phosphatase-like sequences.

Reviewer #2 (Remarks to the Author):

Journal: Nature

Reviewer: Aguirre-Garcia MM Date: 09/23/2023

Title: Acid phosphatases-like proteins, a biogenic amine and leukotriene-binding salivary protein family from the flea *Xenopsylla cheopis*. In this paper the analysis of flea *X. cheopis* salivary acid phosphatases-like proteins lost their expected catalytic activity and became scavengers of biogenic amines and leukotrienes. In a novel contribution regarding the presence of acid phosphatases in flea salivary glands, the authors provide a detailed description of the interaction between recombinant phosphatase proteins and bioactive molecules. I believe this manuscript could be accepted for publication with some suggestions:

Reply: We would like to express our gratitude to the reviewer for dedicating their time and expertise to enhance our manuscript.

Comments on Introduction:

1. Elaborate further on the concepts of phosphatases, including their definition and classification.

Reply: In response to this suggestion, we have enriched the introduction section to provide a more comprehensive overview of the histidine phosphatase superfamily and its existing classification.

2. Provide a more comprehensive account of other studies on the presence of phosphatases in arthropod salivary glands.

Reply: We appreciate the reviewer's enthusiasm for exploring the potential role of phosphatase in the salivary glands of blood-feeding arthropods. However, it's important to note that there is a limited body of research focused on this specific area. Given the considerable functional diversity exhibited by phosphatases and the structural similarity of XcAP-1, -2 and -3 to acid phosphatases, we have briefly expanded the discussion of acid phosphatases and their activity found in the salivary glands of other blood feeding arthropods. We hope that this addition will help set the context for our study and its significance.

Comments on Methods:

It is recommended to supplement your study with some biochemical analyses of the acid phosphatase protein, such as:

1. Determine or mention the protein profile present in the salivary glands and the recombinant proteins by performing SDS-PAGE gel electrophoresis, visualizing the proteins through Coomassie blue staining or silver staining.

Reply: Two comprehensive scientific articles have delved into the composition of *X. cheopis* salivary glands, employing transcriptomic and proteomic methodologies. Furthermore, both studies included SDS-PAGE analysis of flea salivary glands homogenates. In our introduction section, we cited such studies: "**Exploration of the flea salivary gland using transcriptomics and proteomics approaches has revealed its unique composition with an abundance of several acid phosphatase-like sequences lacking one or two catalytic residues (19, 20).**"

2. Characterize the phosphatase proteins in salivary gland extracts and recombinant proteins by using specific inhibitors of their activity, which have been well-established. Analyzing the acid phosphatase activity in the presence of a group of phosphatase inhibitors is suggested.

Reply: As suggested by the reviewer, we have evaluated the phosphatase activity in *X. cheopis* salivary glands homogenates (SGH) in the presence of different inhibitors, including sodium fluoride, sodium tartrate and p-bromolevamisole oxalate. The results of these experiments have been included as **Supplementary figure 2E** of the manuscript. However, it is important to note that, given the absence of the expected enzymatic activity characteristic of members of the acid phosphatase family in the recombinant proteins, we did not perform enzymatic assays in the presence of inhibitors for these proteins.

3. Determine the optimal pH of the generated recombinant proteins to establish whether they are indeed acid phosphatases, as described by the authors.

Reply: In response to the reviewer's suggestion, we have conducted an assessment of the potential enzymatic activity of recombinant XcAP-1, -2 and -3 at various pH levels ranging from 3 to 9. Our findings demonstrated that the recombinant proteins are incapable of processing pNPP across all pH conditions tested. This confirms the absence of the typical catalytic activity present in the members of the acid phosphatase family. The results of these experiments have been incorporated into the manuscript as **Supplementary figure 2A – D**.

Comments on Conclusions:

Propose a potential model for the interaction of acid phosphatase with these inflammatory mediators.

Reply: In accordance with the reviewer's recommendation, we have introduced a "conclusion" section to our manuscript, which succinctly encapsulate our research findings. Additionally, we've integrated a schematic diagram, denoted as **Figure 5**, to illustrate our hypothesis regarding the inhibitory effects of *X. cheopis* salivary acid phosphatase-like proteins on the host's hemostatic and immune responses, ultimately facilitating blood acquisition.

REVIEWERS' COMMENTS:

Reviewer #2 (Remarks to the Author):

I have reviewed the latest version, and agree with the comments and suggestions added. I have no further questions.

I consider that the manuscript is suitable for acceptance.

Reviewer #2 (Remarks to the Author):

I have reviewed the latest version, and agree with the comments and suggestions added. I have no further questions.

I consider that the manuscript is suitable for acceptance.

Reply: We express our gratitude to the reviewer for dedicating time and effort to enhance the overall quality of our manuscript.